# A New Time–Frequency Feature Extraction Method for Action Detection on Artificial Knee by Fractional Fourier Transform

**DOI:** 10.3390/mi10050333

**Published:** 2019-05-20

**Authors:** Tianrun Wang, Ning Liu, Zhong Su, Chao Li

**Affiliations:** 1Beijing Key Laboratory of High Dynamic Navigation Technology, Beijing Information Science and Technology University, Beijing 100101, China; wqwqwq604@hotmail.com (T.W.); sz@bistu.edu.cn (Z.S.); jackbeijing6747@sina.com (C.L.); 2Beijing Institute of Technology, School of Automation, Beijing 100084, China

**Keywords:** action detection, artificial knee, microelectromechanical systems (MEMS), fractional Fourier transform (FRFT)

## Abstract

With the aim of designing an action detection method on artificial knee, a new time–frequency feature extraction method was proposed. The inertial data were extracted periodically using the microelectromechanical systems (MEMS) inertial measurement unit (IMU) on the prosthesis, and the features were extracted from the inertial data after fractional Fourier transform (FRFT). Then, a feature vector composed of eight features was constructed. The transformation results of these features after FRFT with different orders were analyzed, and the dimensions of the feature vector were reduced. The classification effects of different features and different orders are analyzed, according to which order and feature of each sub-classifier were designed. Finally, according to the experiment with the prototype, the method proposed above can reduce the requirements of hardware calculation and has a better classification effect. The accuracies of each sub-classifier are 95.05%, 95.38%, 91.43%, and 89.39%, respectively; the precisions are 78.43%, 98.36%, 98.36%, and 93.41%, respectively; and the recalls are 100%, 93.26%, 86.96%, and 86.68%, respectively.

## 1. Introduction

The knee joint is an important supporting joint of the lower limbs. For a knee joint injury that seriously impedes the movement of the lower limbs, the best treatment is to replace the joint with an artificial joint. Among patients, a large number need amputation, and prosthesis is an important guarantee for the physical and physiological recovery of amputees. In the lower limb prosthesis, the artificial knee joint is the core component and it can help the human body to achieve supported standing and flexible walking so that patients show good gait [1].

Before the 1990s, the movement of prosthetics could not keep pace with the speed of amputees and road conditions, and the movement lacked stability [2]. With the continuous development of bionic intelligent knee prosthesis technology, there are many intelligent knee prostheses, which can achieve dynamic adjustment, real control of gait, and other functions. The intelligent lower limb usually refers to the unpowered intelligent lower limb. The flexion and extension movement of the prosthetic knee joint is driven by the leg stump. The intelligent control system only adjusts the size of the damping torque of the knee joint [3]. For the knee joint of the unpowered intelligent lower limb prosthesis that controls gait by adjusting the damping torque, the accuracy of damping adjustment directly affects the comfort, convenience and safety of the knee joint in question. Generally, the damping adjustment is based on the recognition of the current action state. Among the existing action detection methods of intelligent prosthetics, methods that detect the motion frequency or the position of the piston of a cylinder and the pressure at the bottom of the foot have long lag times and low accuracy. Methods that extract the features of acceleration or angular velocity from the time domain have few features from which to choose, hence they are susceptible to interference and low robustness. So, an accurate, fast, low-cost and stable action detection method is needed.

The action detection often needs to go through the process of data collection, data preprocessing, feature extraction, the establishment of an action detection model, and the use of the model for action detection [4]. Currently, sensors used to collect data can be divided into two major categories of wearable and nonwearable. The nonwearable methods include motion detection methods based on radar [5], and machine vision [6]. The wearable methods include motion detection methods based on accelerometers [7], accelerometers and gyroscopes [8], accelerometers and other sensors [9], and myoelectricity [10]. The nonwearable sensors are not suitable for human motion recognition on prosthesis due to their severe constraints of scene and high cost. In the wearable methods, myoelectricity is inconvenient to wear on a daily basis due to the need to install electrodes in multiple places on the wearer’s legs. The microelectromechanical systems (MEMS) inertial measurement unit (IMU) integrated with accelerometers, gyroscopes and barometers can be integrated into the control circuit and does not require additional operation. So, it is more suitable for human action data acquisition on commercial products.

For feature extraction, the preprocessed data are usually divided into a series of windows by the sliding window mechanism of fixed length first-in, first-out, and the feature values are extracted by a series of methods. Aziz et al., for example, calculated the mean and variance of data in each window to construct 18 dimensional features [11]; Pierleoni et al. discovered that the RMS (Root Mean Square) of acceleration was no less than 2.5 g when the impact process of motion occurs [12]; Cheng et al. judged the state of motion according to an amplitude of acceleration higher than a given threshold [13].

For the establishment of an action detection model, the commonly used methods are the threshold method and the machine learning method. The threshold method determines motion state by comparing the extracted features with a setting threshold; generally, it can be divided into two types: a single state recognition and a multiple state recognition. The machine learning method regards action detection as a typical classification problem and constructs a motion detection model based on the training set composed of various motion data. Typical machine learning algorithms such as support vector machine (SVM) [11,14,15,16], decision tree (DT), Naive Bayes [17], deep learning [18], artificial neural network (ANN) [19], and K nearest neighbor (K-NN) [20] can be used for model construction. The preferable effective algorithms include convolutional neural network (CNN), I support vector machine (1SVM), CNN+1SVM [21], hidden Markova model [22], K-NN and ANN [23].

However, in most action detection methods the machine learning method can achieve higher precision, but it needs lots of features and is a complicated detection model with a slow detection speed and high calculate cost. The threshold method is simple in calculation and fast in speed, which is very suitable for long-term operation on micro-wearable devices with limited resources. However, the selection of its threshold value often depends on the simulated motion data, and there are few features from which to choose. When the distribution of test data and training data is greatly different, it will seriously affect the detection accuracy of the model, so it lacks certain applicability. Therefore, a detection method that can provide sufficient features for selection, at the same time as satisfying high computing speed and low model complexity, is needed. Considering that human action is a time-varying movement, and we cannot only look for features in the time domain or frequency domain, we came up with the idea of using the fractional Fourier transform (FRFT) in order to provide sufficient features from the fractional domain, and also in order to obtain suitable features for low power consumption wearable devices.

Fractional Fourier transform (FRFT) is a representation method on the fractional-order Fourier domain by the signal on the time–frequency plane after the coordinate axis rotates counterclockwise around the origin at any angle. It is a time–frequency analysis method and a generalized Fourier transform. Fractional Fourier transform has many properties that traditional Fourier transform does not have, and is widely applied in scientific research and engineering technology [24]. In this paper, firstly the effect of FRFT is analyzed, and the results of transformation with the same action in different orders, or different actions in the same order, are compared. Then, eight features are selected from the commonly used features to construct a feature vector space. The variation of each feature after FRFT in different orders is analyzed. Then, the dimension of the feature vector is further reduced, and the results of different features after FRFT in different orders are analyzed, according to which we design a classifier. Finally, a prototype of an intelligent knee joint is designed, and the method is verified by experiments.

## 2. Theory

### 2.1. Fractional Fourier Transform (FRFT)

In the field of signal processing, the traditional Fourier transform is a mature and widely used mathematical tool. The fractional order Fourier transform (FRFT) is proposed in the form of pure mathematics by V. Namias from the view of feature and feature function [25]. Then, researchers proposed the concept of FRFT from an optical point of view. It can be proved that these definitions are completely equivalent [26]. FRFT is first applied to optical signal processing because it can be implemented by simple optical devices. In recent years, several rapid algorithms for FRFT have been discovered, so that FRFT has received attention in multiple areas of signal processing.

#### 2.1.1. Definition

Generally, the p-order fractional Fourier transform of a function $f_{p}(u)$ can be expressed as: $f_{p}(u)$ or $F^{p}f(u)$, where, $F^{p}f(u)$ can be interpreted as the operator $F^{p}$ on the function f(u) whose result is in the domain u.

The definition of the fractional Fourier transform [27] is
(1)$$f_{p}(u)=\int_{-\infty }^{+\infty }K_{p}(u,t)f(t)dt$$
where $K_{p}(u,t)=\{\begin{array}{l} A_{\alpha }\exp[j\pi (u^{2}\cot\alpha -2ut\csc\alpha +t^{2}\cot\alpha )],\alpha \neq n\pi \\ \delta (u-t),\alpha =2n\pi \\ \delta (u+t),\alpha =(2n+1)\pi \end{array}$ is the kernel function of the fractional Fourier transform, $A_{\alpha }=\frac{\exp[-j\pi \mathrm{sgn}(\sin\alpha )/4+j\alpha /2]}{{\left|\sin\alpha \right|}^{1/2}}$, $\alpha =\frac{p\pi }{2}$, n is integer.

After sorting, it is shown as follows:(2)$$f_{\alpha }(u)=A_{\alpha }T_{t}(u)\int_{-\infty }^{+\infty }T_{s}(u-x)[T_{t}(x)f(x)]dx$$
where $T_{t}(x)=\exp(-j\pi tx^{2})$, t=tan(α/2), s=−csc(α).

Noticing that, $F^{4n}$ and $F^{4n\pm 2}$ are equivalent to identity operator τ and parity operator P respectively. For p=1, there are $\alpha =\frac{\pi }{2}$, $A_{a}=1$ and $f_{1}(u)=\int_{-\infty }^{+\infty }e^{-j2\pi ut}f(t)dt$.

Apparently, $f_{1}(u)$ is the Fourier transformation of f(u). The 0-order transformation is defined as the function itself, and the definition by p or α is periodic in 4 or 2π due to $\alpha =\frac{p\pi }{2}$ only appearing in the parameter position of the trig function.

Except for comparison with the time domain or the frequency domain, to explain the reason that choose FRFT to extract features, Li Chao derived the relationship between the amplitude mean
$\bar{A}$
in the time domain and the amplitude *B* in the fractional domain by the timewidth–bandwidth in theory [28].

From the fraction domain sampling theorem, the relationship between the fraction domain bandwidth Bu and the frequency domain bandwidth *B* is:(3)Bu=Bsinα
where α is the rotation angle of the fraction domain.

The timewidth and bandwidth of the fraction domain signal are defined as follows:(4)$$\Delta t^{2}≜{\int_{-\infty }^{+\infty }\left|\left(t-t_{0}\right)x(t)\right|}^{2}dt$$
(5)$$\Delta u_{a}{}^{2}≜{\int_{-\infty }^{+\infty }\left|\left(u_{a}-u_{a}{}_{0}\right)x(u_{a})\right|}^{2}du_{a}$$
where $u_{a}$ is the frequency of the fractional domain and $x(u_{a})$ is the FRFT of x(t).

From the uncertainty principle of the signal frequency domain:(6)$$\Delta t^{2}\cdot \Delta u^{2}\geq \frac{1}{4}\mathrm{is}(\Delta t\cdot \Delta u\geq \frac{1}{2})$$
where Δt is the timewidth and Δu is the bandwidth.

The uncertainty principle of the fractional domain can be derived from the three Equations (4)–(6):(7)$$\Delta t^{2}\cdot \Delta u_{a}{}^{2}\geq \frac{\sin^{2}\alpha }{4}is(\Delta t\cdot \Delta u_{a}\geq \frac{\sin\alpha }{2})$$

From Parseval, the energy of the signal in the time domain is the same as that in the fractional domain:(8)$$E=E_{\alpha }$$
where E is the total energy in the time domain, and $E_{\alpha }$ is the total energy in fraction domain. Furthermore:(9)$$E=\frac{1}{2}\bar{A}\cdot \Delta t^{2},E_{\alpha }=\frac{1}{2}\bar{B}\cdot \Delta u_{\alpha }{}^{2}$$

Combining Equations (7)–(9), it can be observed that the signal’s amplitude average $\bar{A}$ in the time domain and the $\bar{B}$ of the signal show a non-linear change, and it is related to the changed angle in the fractional domain [29,30].

Therefore, a sequence of values with no significant difference in amplitude in time domain can be converted into a fractional sequence with significant differences in amplitude by choosing an appropriate FRFT order [28].

#### 2.1.2. Discrete FRFT

Equation (2) is the calculation method in the continuous domain. Such continuous transformation cannot be calculated in practice, and it is usually necessary to carry out numerical calculation by sampling and interpolating continuous signals. According to the Shannon interpolation equation and numerical integration operation, the discrete calculation equation of FRFT is shown in Equation (10):(10)$$f_{\alpha }(\frac{k}{2\Delta })\approx \frac{A_{\alpha }}{2\Delta }\exp[-j\pi \tan(\alpha /2){\left(\frac{k}{2\Delta }\right)}^{2}]\times \sum_{l=-N}^{N-1}\exp[j\pi {\left(\frac{k-l}{2\Delta }\right)}^{2}\csc\alpha ]\left\{\exp[-j\pi {\left(\frac{l}{2\Delta }\right)}^{2}\tan(\alpha /2)f(\frac{l}{2\Delta })]\right\}$$

After sorting, it is shown as follows:(11)$$f_{\alpha }(x_{k})\approx \frac{x_{k}C_{\alpha }}{k}E_{t}(u_{k})\sum_{l=-N}^{N-1}\left\{E_{s}(x_{k-l})[E_{t}(x_{l})f(x_{l})]\right\}$$
where $E_{t}(u_{k})=\exp(-j\pi tu_{k}^{2})$, t=tan(α/2), s=−csc(α) and $x_{k}=u_{k}=k/2\Delta$. Δ is the dimension normalized time domain or frequency domain scale.

#### 2.1.3. Transform Effect

According to the theory above, the inertial data collected by MEMS can be processed by FRFT. In contrast to the time domain and the frequency domain, the fractional domain has a better diversity and can provide more optional features. Figure 1 and Figure 2 are the results of the same data after FRFT in different orders. Figure 3a is the results of the same class data after FRFT in the same order. Figure 3b is the results of different data after FRFT in the same order. It can be observed that the results of the same data after FRFT in different orders show significant differences, and the results of different data after FRFT in the same order also show differences. Therefore, different actions can be distinguished by selecting an appropriate order and an appropriate feature.

### 2.2. Feature

After the performance of FRFT on the collected data, the appropriate features should be extracted and then classified according to the features. The commonly used features are shown in Table 1. However, not all the features are appropriate for the data in MEMS IMU, besides, too many features will lead to excessive computing costs, time costs, storage costs, and reduce the endurance of the device. However, too few features will not reach enough accuracy to classify different actions. Therefore, it is necessary to select appropriate features for action recognition.

In this paper, the extreme difference range, standard deviation std, variance var, interquartile range IQR, mean mean, mean of peaks pksMean, and number of peaks pksNum were selected to form the feature_vector:(12)feature_vector=[range,std,var,rms,IQR,mean,pksMean,PksNum]

Figure 4, Figure 5, Figure 6, Figure 7, Figure 8, Figure 9, Figure 10 and Figure 11 are the analysis of each feature during the order of FRFT changed from 0 to 1. Figure 4, Figure 6, Figure 8 and Figure 10 are the FRFT results of the eight features of four actions, namely walk, run, upstairs, and dwstairs (downstairs), in different orders. Figure 5, Figure 7, Figure 9 and Figure 11 are the standard deviation and mean of eight features of each action. Table 2, Table 3, Table 4 and Table 5 are the mean, minimum standard deviation and corresponding order of standard deviation of each feature of each action. It can be observed that, except for rms, all the other features change significantly, but this does not mean that rms cannot be used for classification. Besides, it can be observed that it might be useful to classify walk and dwstairs by rms and mean, to classify upstairs and dwstairs by range and pksNum, and to classify walk, dwstairs and run by std and pksNum.

Different features have different classification effects for the inertia data in MEMS IMU. Figure 12 shows the effect of distinguishing two actions by two kinds of features. It can be observed that to classify walk and dwstairs, the combination of pksNum and std is better than the combination of range and pksMean, and the order = 0.67 is better than order = 0.20. To classify upstairs and dwstairs, the combination of range and pksNum is better than the combination of rms and IQR, and the order = 0.64 is better than order = 0.20. To classify walk and upstairs, the combination of rms and mean is better than the combination of std and IQR, and the order = 0.71 is better than order = 0.20. To separate run from the other actions, the combination of rms and mean is better than the combination of range and IQR, and it is better when order = 0.75. Figure 13 shows the effect of classification with two features extracted directly in the time domain without FRFT. It can be observed that the effect was not good for classification.

### 2.3. Classifier

According to the conclusion above, different features and different orders have different effects for classification. The optimal order and feature of each action are different. A binary classifier was designed which classifies by changing the order and feature value of each sub-classifier. Only one action is separated at a time, and then all motion gestures are recognized by multiple classifications. The structure of the classifier is shown in Figure 14.

## 3. Experiment

### 3.1. Experiment Design

According to the conclusion above, an experiment for verification is designed. The information of the subject is shown in Table 6. To protect the privacy of amputees, the photos of amputees will not be shown here. The artificial knee used in the experiment is shown in Figure 15, the MEMS IMU used in the experiment is shown in Figure 16, and the main specifications of the devices are given in Table 7.

### 3.2. Results and Analysis

After the experimental data collection, the same processing method is used to extract the feature vector through FRFT after dividing the cycle, and the classifier designed above is used for classification, the results are shown in Figure 17 and the performance of the sub-classifier is detailed in Table 8. The comparison of the accuracy in fractional domain and time domain is in Table 9. It can be seen that the effect of fractional domain is better than the effect of time domain.

## 4. Conclusions

With the aim of designing an action detection method on artificial knee, a new time–frequency feature extraction method is proposed. This method is targeted at four common actions of the artificial knee wearer, and extracted features from the inertia data are measured by MEMS IMU, using fractional Fourier transform (FRFT) to magnify the diversity of features. FRFT is employed to extract the appropriate feature vectors and construct a feature vector composed of eight features.

By analyzing the results of these features after FRFT in different orders, it was discovered that it may have a good effect to classify walk and dwstairs by rms and mean, to classify upstairs and dwstairs by range and pksNum, and to classify walk, dwstairs and run by std and pksNum.

The results of the classification with different features and orders are also analyzed. It was discovered that in order to classify walk and dwstairs, the combination of pksNum and std is better than the combination of range and pksMean, and it is better when order = 0.67 than order = 0.20. To classify upstairs and dwstairs, the combination of range and pksNum is better than the combination of rms and IQR, and it is better when order = 0.64 than order = 0.20. To classify walk and upstairs, the combination of rms and mean is better than the combination of std and IQR, and it is better when order = 0.71 than order = 0.20. To separate run from the other actions, the combination of rms and mean is better than the combination of range and IQR, and it is better when order = 0.75.

Finally, the verification experiment is designed with the artificial lower limb knee prototype. The results show that the method that we proposed has a good classification performance while also reducing the requirements of hardware calculation. The accuracies of each sub-classifier are 95.05%, 95.38%, 91.43%, and 89.39%, respectively; the precisions are 78.43%, 98.36%, 98.36%, and 93.41%, respectively; and the recalls are 100%, 93.26%, 86.96%, and 86.68%, respectively.

## Figures and Tables

**Figure 1 micromachines-10-00333-f001:**
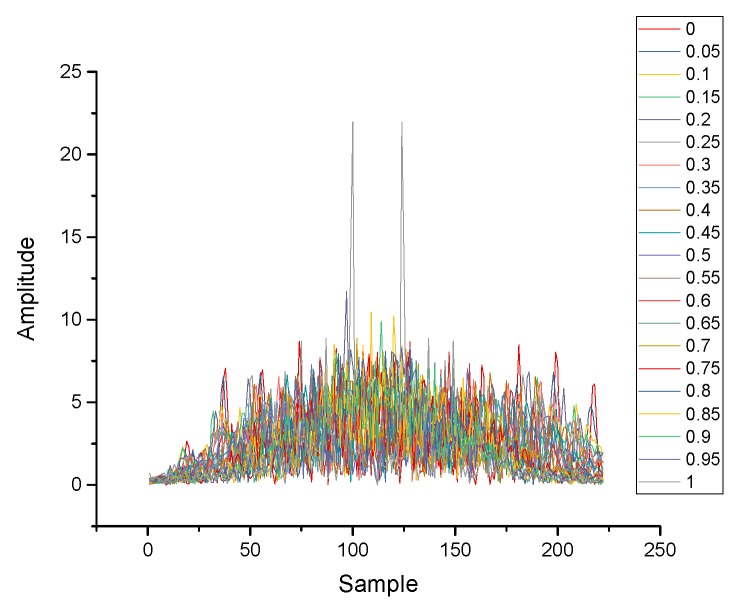
Results of Multiorder fractional Fourier transform (FRFT) with single action (order: 0–1).

**Figure 2 micromachines-10-00333-f002:**
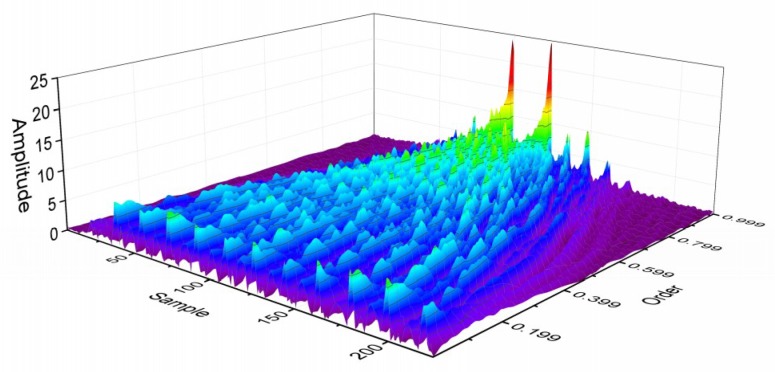
Results of Multiorder FRFT with single action in 3D (order: 0–1).

**Figure 3 micromachines-10-00333-f003:**
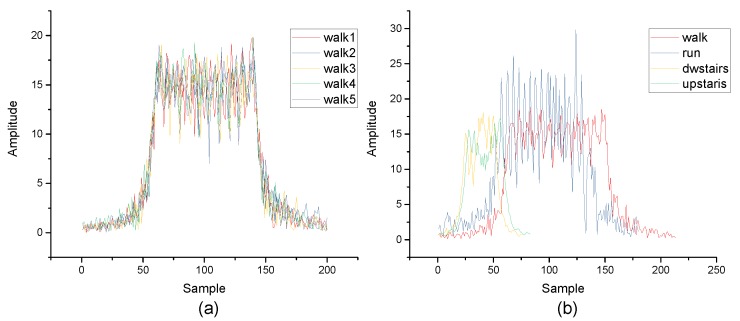
(**a**) The results of FRFT of walking in 0.7 order. (**b**) The results of FRFT of different actions in 0.7 order.

**Figure 4 micromachines-10-00333-f004:**
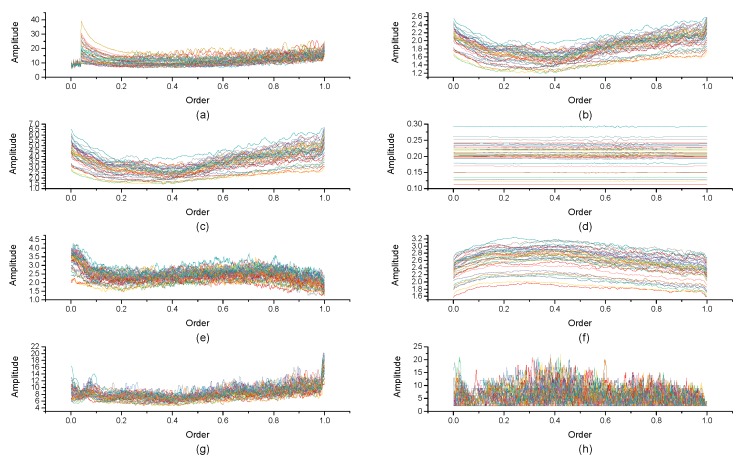
The Order–Amplitude figures of different features of 40 groups for the walk action. (**a**) range, (**b**) std, (**c**) var, (**d**) rms, (**e**) IQR, (**f**) mean, (**g**) pksMean and (**h**) pksNum.

**Figure 5 micromachines-10-00333-f005:**
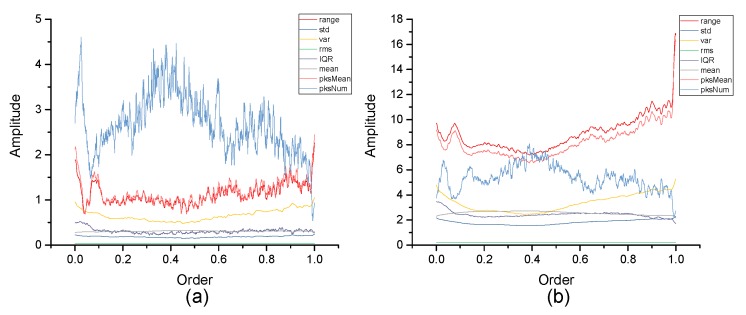
Standard deviation and mean of eight features of the walk action. (**a**) The standard deviation of each feature, (**b**) The mean of each feature.

**Figure 6 micromachines-10-00333-f006:**
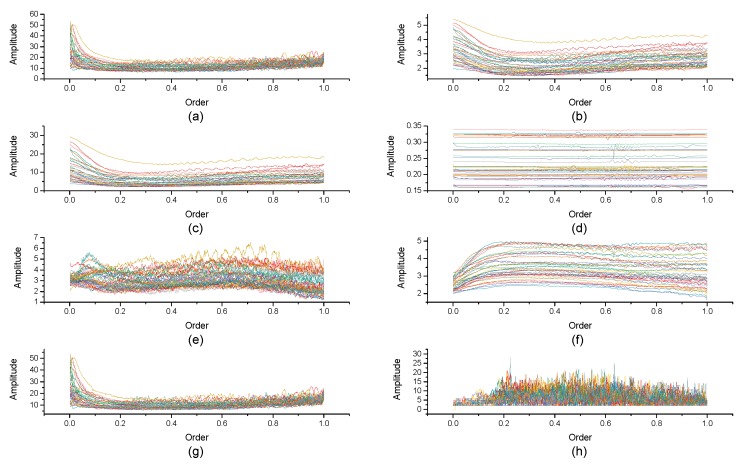
The Order–Amplitude figures of different features of 40 groups for the upstairs action. (**a**) range, (**b**) std, (**c**) var, (**d**) rms, (**e**) IQR, (**f**) mean, (**g**) pksMean and (**h**) pksNum.

**Figure 7 micromachines-10-00333-f007:**
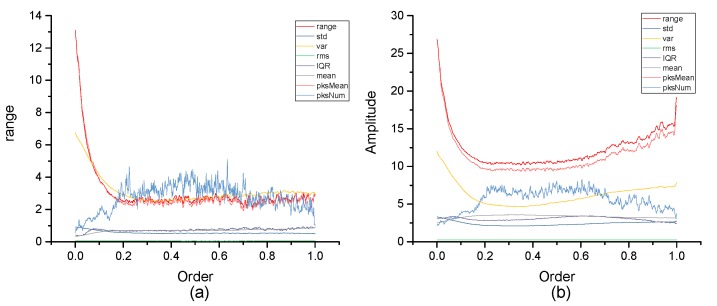
Standard deviation and mean of eight features of the upstairs action. (**a**) The standard deviation of each feature, (**b**) The mean of each feature.

**Figure 8 micromachines-10-00333-f008:**
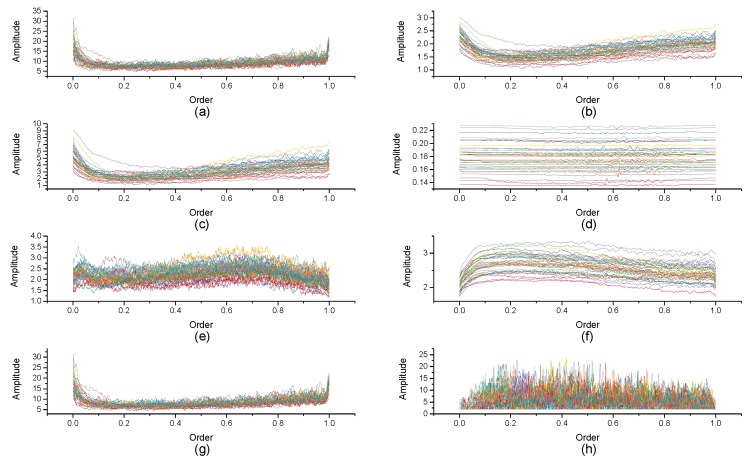
The Order–Amplitude figures of different features of 40 groups for the dwstairs action. (**a**) range, (**b**) std, (**c**) var, (**d**) rms, (**e**) IQR, (**f**) mean, (**g**) pksMean and (**h**) pksNum.

**Figure 9 micromachines-10-00333-f009:**
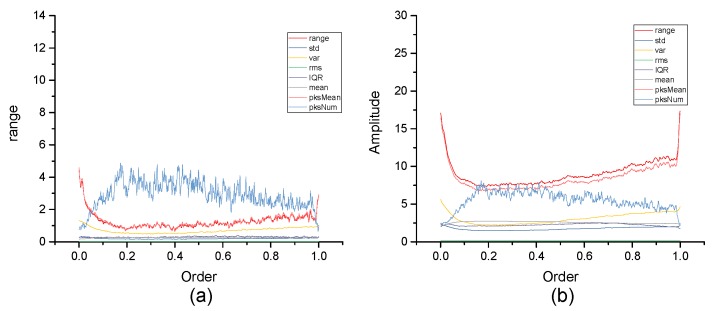
Standard deviation and mean of eight features of the dwstairs action. (**a**) The standard deviation of each feature, (**b**) The mean of each feature.

**Figure 10 micromachines-10-00333-f010:**
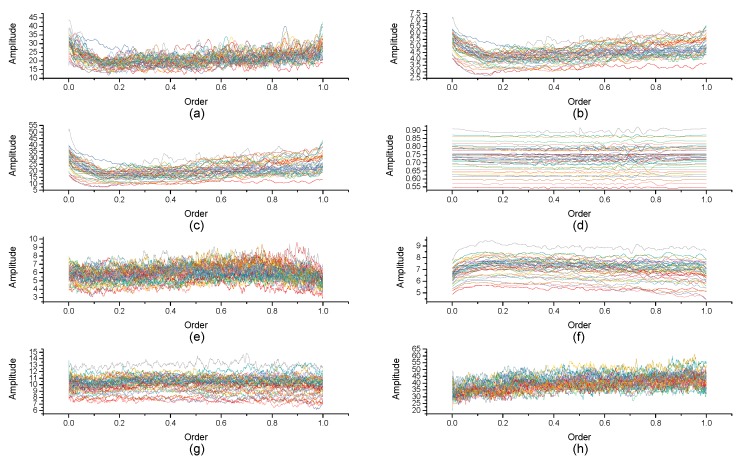
The Order–Amplitude figure of different features of 40 groups for the run action. (**a**) range, (**b**) std, (**c**) var, (**d**) rms, (**e**) IQR, (**f**) mean, (**g**) pksMean and (**h**) pksNum.

**Figure 11 micromachines-10-00333-f011:**
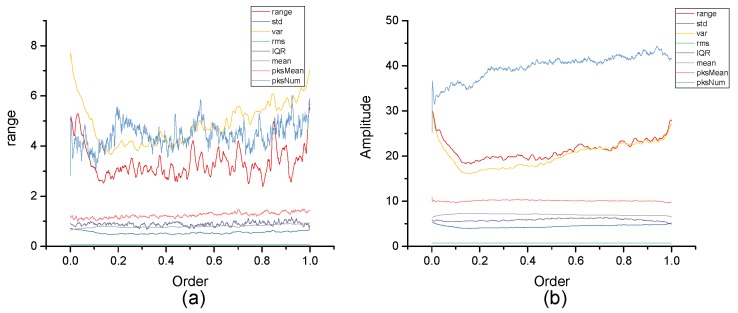
Standard deviation and mean of eight features of the run action. (**a**) The standard deviation of each feature, (**b**) The mean of each feature.

**Figure 12 micromachines-10-00333-f012:**
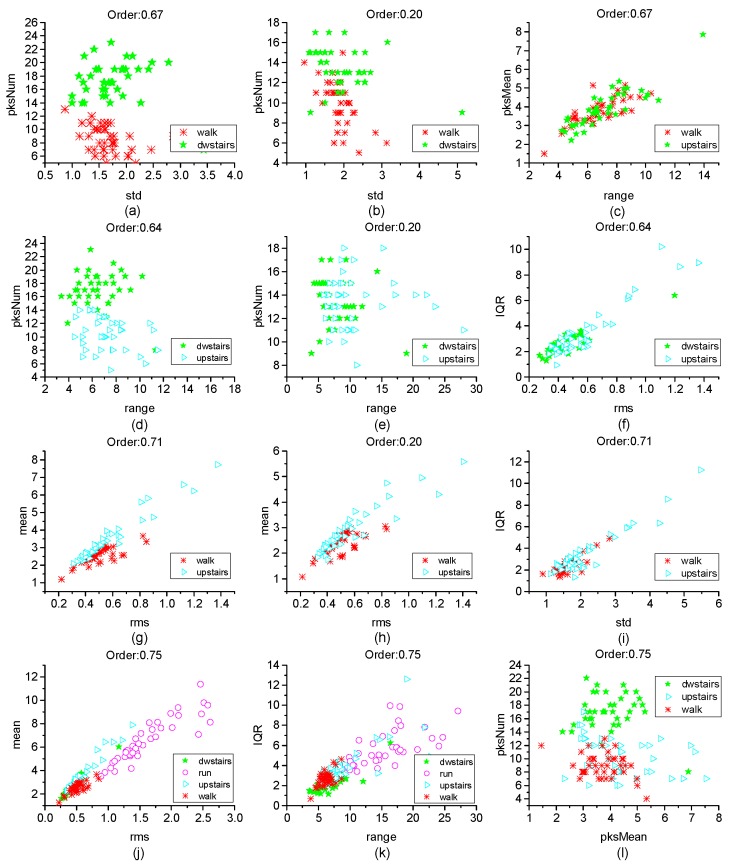
The effect of classification. (**a**) walk and dwstairs with the features std and pksNum when Order = 0.67; (**b**) walk and dwstairs with the features std and pksNum when Order = 0.20; (**c**) walk and dwstairs with the features range and pksMean when Order = 0.67; (**d**) dwstairs and upstairs with the features range and pksNum when Order = 0.64; (**e**) dwstairs and upstairs with the features range and pksNum when Order = 0.20; (**f**) dwstairs and upstairs with the features rms and IQR when Order = 0.64; (**g**) walk and upstairs with the features rms and mean when Order = 0.71; (**h**) walk and upstairs with the features rms and mean when Order = 0.20; (**i**) walk and upstairs with the features std and IQR when Order = 0.71; (**j**) all the actions with the features rms and mean when Order = 0.75; (**k**) all the actions with the features range and IQR when Order = 0.75; (**l**) all the actions with the features pksMean and pksNum when Order = 0.75.

**Figure 13 micromachines-10-00333-f013:**
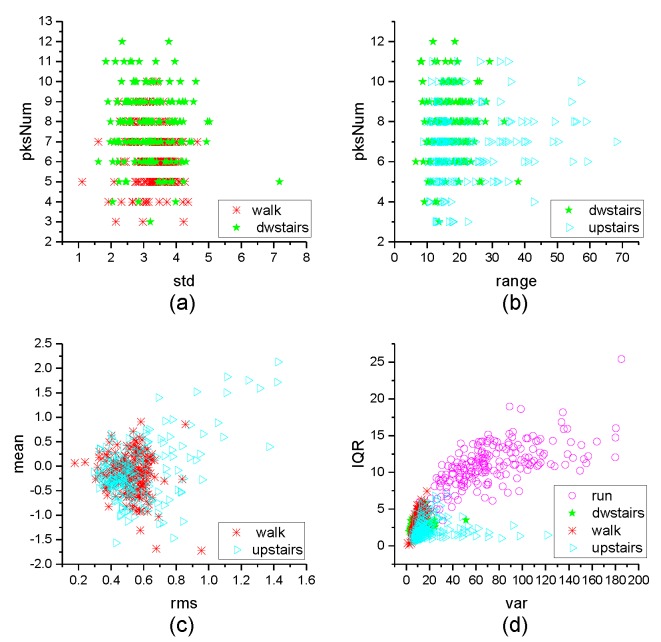
The effect of classification with two features extracted directly in the time domain without FRFT. (**a**) walk and dwstairs with the features std and pksNum; (**b**) dwstairs and upstairs with the features range and pksNum; (**c**) walk and upstairs with the features rms and mean; (**d**) all the actions with the features var and IQR.

**Figure 14 micromachines-10-00333-f014:**
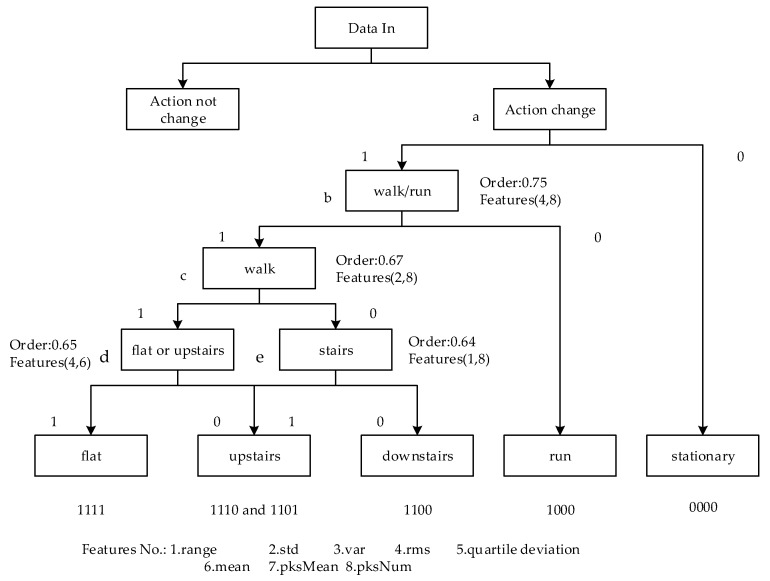
The structure and process of the classifier. As different orders and feature vectors are needed to classify walk, upstairs and dwstairs, one can distinguish walk and dwstairs first, and then separate upstairs from walk and dwstairs, respectively, then separate upstairs from walk and dwstairs, respectively.

**Figure 15 micromachines-10-00333-f015:**
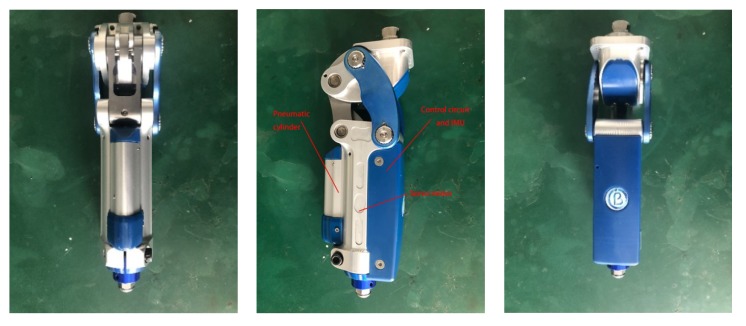
The artificial knee. The position of the pneumatic cylinder, servo motor and control circuit with inertial measurement unit (IMU) have been pointed out.

**Figure 16 micromachines-10-00333-f016:**
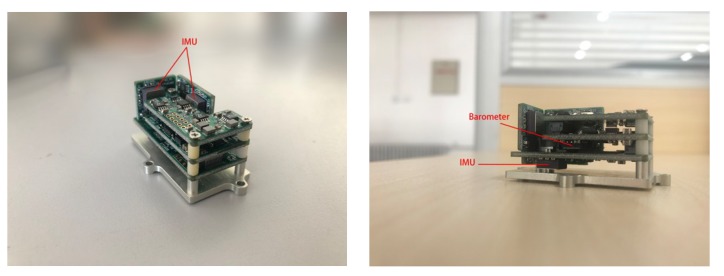
MEMS (microelectromechanical systems)-IMU, the position of IMU and barometer have been pointed out.

**Figure 17 micromachines-10-00333-f017:**
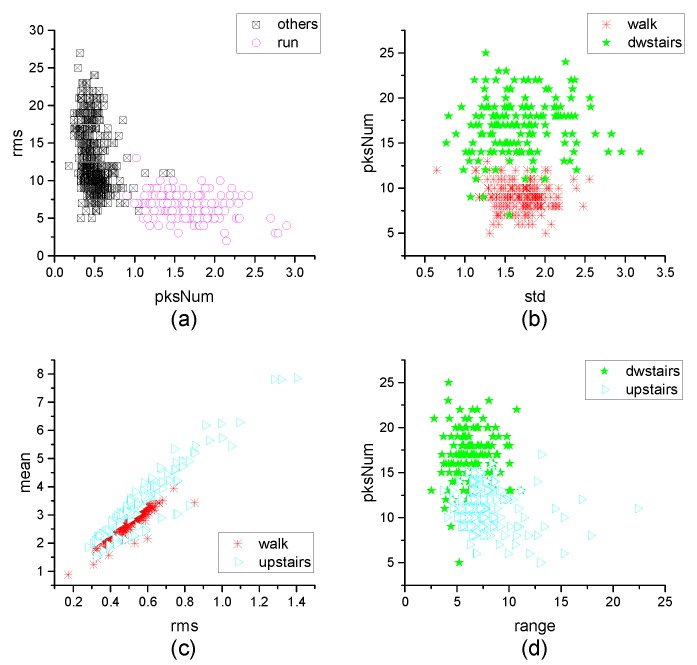
The results of each sub-classifier. (**a**) run and other actions with the features rms and pksNum; (**b**) walk and dwstairs with the features std and pksNum; (**c**) walk and upstairs with the features rms and mean; (**d**) dwstairs and upstairs with the features range and pksNum.

**Table 1 micromachines-10-00333-t001:** The commonly used features and the calculation method.

Features	Calculation Method
Maximum	max(A)
Minimum	min(A)
Mean	$mean(A)=\sum_{i}\frac{A_{i}}{n}$
Extreme Difference	range(A)=max(A)−min(A)
Variance	$var(A)=\frac{\sum_{i}{\left(A_{i}-mean(A)\right)}^{2}}{n}$
Standard Deviation	$std(A)=\sqrt{var(A)}$
Root Mean Square	$rms(A)=\sqrt{\frac{\sum_{i}A_{i}^{2}}{n}}$
Absolute Value	$abs(A_{i})=\left|A_{i}\right|$
Signal Amplitude Area	$sma(A)=\frac{1}{t}(\int_{0}^{t}\left|A_{x}\right|dt+\int_{0}^{t}\left|A_{y}\right|dt+\int_{0}^{t}\left|A_{z}\right|dt)$
Correlation Coefficient	$cc(A)=\frac{cov(A_{x},A_{y})}{var(A_{x},A_{y})}$
Interquartile Range	$IQR=Q_{3}-Q_{1}$
Number of Peaks	The number of peaks in signal (pksNum)
Mean of Peaks	$pksMean=\frac{\sum_{i}pks_{i}}{n}$

**Table 2 micromachines-10-00333-t002:** Mean and min of standard deviation of each feature of walk.

Feature	STD		
	Min		Mean
	Value	Order	
Range	0.6996	0.390	1.1189
Standard Deviation (Std)	0.1494	0.453	0.1854
Variance (Var)	0.4785	0.453	0.6612
Root Mean Square (RMS)	0.0382	0.635	0.0387
Interquartile Range (IQR)	0.2171	0.310	0.3105
Mean	0.2799	1	0.3076
Mean of Peaks (pksMean)	0.6857	0.043	1.1725
Number of Peaks (pksNum)	0.5430	0.992	2.6806

**Table 3 micromachines-10-00333-t003:** Mean and min of standard deviation of each feature of upstairs.

Feature	STD		
	Min		Mean
	Value	Order	
Range	1.9913	0.981	3.0821
Std	0.4898	0.997	0.5659
Var	2.4833	0.428	3.1291
RMS	0.0527	0.607	0.0535
IQR	0.3743	0.130	0.7365
Mean	0.3249	1	0.7060
pksMean	1.9208	0.701	2.8684
pksNum	0.5633	1	2.7401

**Table 4 micromachines-10-00333-t004:** Mean and min of standard deviation of each feature of dwstairs.

Feature	STD		
	Min		Mean
	Value	Order	
Range	0.6453	0.400	1.2644
Std	0.1589	0.281	0.1978
Var	0.4816	0.308	0.7028
RMS	0.0209	0.624	0.0213
IQR	0.2369	0.212	0.3018
Mean	0.1681	1	0.2500
pksMean	0.6681	0.194	1.3048
pksNum	0.6485	0.997	2.9343

**Table 5 micromachines-10-00333-t005:** Mean and min of standard deviation of each feature of run.

Feature	STD		
	Min		Mean
	Value	Order	
Range	2.3835	0.803	3.3444
Std	0.4612	0.168	0.5488
Var	3.6646	0.169	4.9012
RMS	0.0783	0.663	0.0803
IQR	0.6780	0.318	0.8944
Mean	0.6471	0.020	0.8000
pksMean	0.9936	0.031	1.2464
pksNum	2.8274	0.991	4.4601

**Table 6 micromachines-10-00333-t006:** Information of the subject.

Project	Value
Age	24
Weight	65 kg
Position of prosthesis	Left
High	174 cm
Gender	M

**Table 7 micromachines-10-00333-t007:** Main specifications of devices.

Parameters		Main Specifications		
Sustainable Working Hours		≥ 24 h		
Operating Voltage		3.3 to 30 V		
Sustainable Working Temperature		−40 °C to 85 °C		
Power Consumption		550 mW@5.0 V		
Core Circuit Board Dimensions		33 mm × 20 mm × 22 mm		
Weight		<100 g		
Sensing Range		0° to 360°		
Static Accuracy		±0.5° (roll, pitch); ±1° (yaw)		
Dynamic Accuracy		±1° RMS		
Resolution		0.05°		
Output Frequency		0.01 to 100 Hz		
**Sensor**	**Accelerometer**	**Gyroscope**	**Parameters**	**Barometer**
Measure range	±10 g	±1000°/s	Measure range	10 to 1200 mbar
Nonlinear	<0.2% of FS	<0.1% of FS	Resolution	10 cm
Bias stability	±4 mg	9.2°/h	Bias stability	±1 mbar/year

**Table 8 micromachines-10-00333-t008:** Performance of each sub-classifier.

Sub-Classifier	b	c	d	e
Accuracy	0.9505	0.9538	0.9143	0.8939
Precision	0.7843	0.9836	0.9836	0.9341
Recall	1	0.9326	0.8696	0.8667
F1–Score	0.3053	0.7018	0.7157	0.6753

**Table 9 micromachines-10-00333-t009:** Accuracy of each sub-classifier in fractional domain and time domain.

Sub-Classifier	Fractional Domain	Time Domain
b	0.9505	0.9011
c	0.9538	0.8259
d	0.9143	0.7424
e	0.8939	0.7629

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
