# Peer review of "A New Time–Frequency Feature Extraction Method for Action Detection on Artificial Knee by Fractional Fourier Transform"

_micromachines, 2019, doi:10.3390/mi10050333_

Round 1

Reviewer 1 Report

The paper would be of solid interest to the readership. The manuscript suffers from numerous english usage and grammatical errors.

Author Response

To Micromachines:

Dear Editor(s) / Reviewer,

Thank you for your letter and for the reviewer’s comments concerning our manuscript entitled "A New Time-Frequency Feature Extraction Method for Action Detection on Artificial Knee by Fractional Fourier Transform ". Those comments are all valuable and very helpful for revising and improving our paper, as well as the important guiding significance to our researches. We have studied comments carefully and have made correction which we hope meet with approval. The main corrections in the paper are as flowing:

1. Revised English usage and grammatical errors.

2. Supplemented the comparison of classify effect between the method that extract features after FRFT and the method that extract features from time domain directly.

3. Revised the reference.

The responds to the reviewer’s comments are as flowing:

Reviewer # 1:

1.Response to comment:The manuscript suffers from numerous English usage and grammatical errors.

Response: Revised it.

We appreciate for Editors/Reviewer’s warm work earnestly, and hope that the correction will meet with approval.

Once again, thank you very much for your comments and suggestions.

Sincerely Yours,

Ning Liu

Reviewer 2 Report

Paper needs improvement, I specified focus on two things. 

They should compare performance with other methods. 

Proper referencing. 

Author Response

To Micromachines:

Dear Editor(s) / Reviewer,

Thank you for your letter and for the reviewer’s comments concerning our manuscript entitled "A New Time-Frequency Feature Extraction Method for Action Detection on Artificial Knee by Fractional Fourier Transform ". Those comments are all valuable and very helpful for revising and improving our paper, as well as the important guiding significance to our researches. We have studied comments carefully and have made correction which we hope meet with approval. The main corrections in the paper are as flowing:

1. Revised English usage and grammatical errors.

2. Supplemented the comparison of classify effect between the method that extract features after FRFT and the method that extract features from time domain directly.

3. Revised the reference.

The responds to the reviewer’s comments are as flowing:

Reviewer # 2:

1.Response to comment:They should compare performance with other methods.

Response: Supplemented the comparison of classify effect between the method that extract features after FRFT and the method that extract features from time domain directly. Figure 13 is the effect of classification with 2 features extracted directly in time domain without FRFT, and Table 9 is the accuracy of each sub-classifier in fractional domain and time domain

2.Response to comment:Proper referencing.

Response: References [15-18] are supplemented.

We tried our best to improve the manuscript and made some changes in the manuscript. These changes will not influence the content and framework of the paper. And here we did not list the changes but marked with “Track Changes”.

We appreciate for Editors/Reviewer’s warm work earnestly, and hope that the correction will meet with approval.

Once again, thank you very much for your comments and suggestions.

Sincerely Yours,

Ning Liu
